# FEW-SHOT ANOMALY DETECTION ON INDUSTRIAL IMAGES THROUGH CONTRASTIVE FINE-TUNING

## ABSTRACT

Detecting abnormal products through imagery data is essential to quality control in manufacturing. Existing approaches towards anomaly detection (AD) often rely on substantial amount of anomaly-free samples to train representation and density models. Nevertheless, large anomaly-free datasets may not always be available before inference stage and this requires building an anomaly detection framework with only a handful of normal samples, a.k.a. few-shot anomaly detection (FSAD). We propose two techniques to address the challenges in FSAD. First, we employ a model pretrained on large source dataset to initialize model weights. To ameliorate the covariate shift between source and target domains, we adopt contrastive training on the few-shot target domain data. Second, to encourage learning representations suitable for downstream AD, we further incorporate cross-instance pairs to increase tightness within normal sample cluster and better separation between normal and synthesized negative samples. Extensive evaluations on six few-shot anomaly detection benchmarks demonstrate the effectiveness of the proposed method.

## 1 INTRODUCTION

Industrial defect detection is an important real-world use-case for visual anomaly detection methods. In this setting, anomaly detection models typically have to be trained with only defect-free, or normal images, as defects rarely occur on functioning production lines. Anomaly detection methods for this one-class classification setting typically assume that normal images are available in abundance, even though this may not always be the case. For example, in certain applications such as semiconductor manufacturing where image acquisition requires 3D scans using specialized equipment (Pahwa et al., 2021), acquiring defect-free images is time-consuming and costly. Flexible manufacturing systems also require rapid adaptation to changes in the type and quantity of products to be manufactured (Shivanand, 2006). As a result, large numbers of defect-free images may not be available for new products, or in the initial stages of bootstrapping a visual inspection system.

Although anomaly detection in general is a well-studied topic (Chandola et al., 2009; Pang et al., 2021b), anomaly detection on images with only few normal and no abnormal images, or few-shot anomaly detection (FSAD), has only recently begun to receive attention from the community (Sheynin et al., 2021; Huang et al., 2022). In their pioneering work, Sheynin et al. (2021) developed a generative adversarial model to distinguish transformed image patches from generated ones. However, such adversarial models may be tricky to tune (Kodali et al., 2017) and the method requires multiple transformations on test samples at inference time, resulting in additional computation overhead. The more recent work of Huang et al. (2022) learns a common model over multiple classes of normal images using a feature registration proxy task, but their method requires a training set with normal images from multiple known classes, which is a more restrictive setting.

In this work, we develop a simple yet effective method for few-shot anomaly detection. We achieve this by synergistically combining transfer learning from a pretrained model with representation learning on the few-shot normal data. Finetuning from a backbone network pretrained on a large source domain dataset, e.g. ImageNet (Russakovsky et al., 2015), allows reusing good low-level feature extractors and better initialization of network parameters (Kornblith et al., 2019). We believe finetuning from pretrained weights could particularly contribute to few-shot anomaly detection when not enough training data is available for training good representations. However, as pointed

out by some existing work (Xu et al., 2022; Li et al., 2021b), directly reusing the pretrained weights may not fully unleash the power of finetuning. This is probably caused by two factors. First, when the source domain data has a different data distribution from the target domain, the covariate shift (Wang & Deng, 2018) causes performance degradation. Second, due to the fact that anomaly detection requires feature representation that separates normal samples from abnormal ones. The representations learned from ImageNet pretraining tasks, mostly semantic image classification, is not necessarily optimal for anomaly detection.

To ameliorate the covariate shift between source and target domain data, we first propose to introduce contrastive training to adapt pretrained model weights to the target data distribution for downstream anomaly detection. Given initial model weights, we optimize a contrastive loss defined on all available few-shot normal examples so that the pretrained low-level features will be adjusted towards the target data distribution. We further encourage learnt feature representations to be suited to the downstream anomaly detection task by encouraging normal samples to form a cluster in feature space. To achieve this, we introduce a cross-instance positive pair loss that randomly samples two normal samples and encourages their feature embeddings to be close. Note that this differs from standard contrastive training as closeness is encouraged across two *different* normal samples instead of a sample and its augmented version. Finally, when prior knowledge on the anomalies is available, e.g. we are able to synthesize negative examples (Li et al., 2021a), we further introduce an additional negative pair loss to encourage better separation between normal and synthesized anomalous examples. We empirically reveal that the choice of negative sample synthesis is crucial to the success of FSAD and should be used only when concrete prior knowledge on the anomalies is available. We summarize the contribution of this work as below,

- We approach anomaly detection for industrial defect inspection from a transfer learning perspective. We propose to do contrastive training on few-shot normal samples in the target domain to alleviate the distribution shift between source and target domains.

- We further introduce an across instance positive pair loss to encourage normal samples to form a tight cluster in the embedding space for better density-based anomaly detection.

- When prior knowledge on negative sample is available a negative pair loss is further incorporated to allow better separation between normal and synthesized negative samples.

- We demonstrate superior performance on 4 real-world industrial defect identification datasets and 2 synthetic corruption identification datasets.

## 2 RELATED WORK

**Anomaly Detection:** Traditional anomaly detection (AD) methods include PCA, cluster analysis (Kim & Scott, 2012) and one-class classification (Schölkopf et al., 2001). With the advent of deep learning, representation learning is employed to avoid manual feature engineering and kernel construction. This leads to novel anomaly detection methods based on generative adversarial networks (GAN) (Perera et al., 2019; Schlegl et al., 2017) and Autoencoders (Bergmann et al., 2019a). Among them, anoGAN (Schlegl et al., 2017) was proposed to learn the manifold of normal samples and anomalous samples cannot be perfectly projected onto the normal manifold by the generator learned solely with normal samples. However, it requires expensive optimization for detecting abnormal samples and training GANs is prone to some well-known challenges including instability and mode collapse. Among the autoencoder based approaches, (Bergmann et al., 2019a) adopted SSIM metric as the similarity measure between input and reconstructed images. Recently an effective line of works approach AD through representation learning and formulate AD as detecting outliers in the learned representation space (Ruff et al., 2018; Golan & El-Yaniv, 2018; Sohn et al., 2021). Among these works, deep SVDD (Ruff et al., 2018) proposed to learn a feature embedding that groups normal samples closer to a cluster center. Follow-up works develop self-supervised pretraining methods to learn representations suitable for separating abnormal samples from normal ones by optimizing a proxy task (Golan & El-Yaniv, 2018; Sohn et al., 2021; Li et al., 2021a). Anomaly detection is then implemented through fitting a density model on the learnt representations of normal training samples. These approaches prevail in many anomaly detection benchmarks and are computationally efficient. Nevertheless, representation learning requires a substantial amount of training data which may not be readily available in certain industrial environments.

**Few-Shot Anomaly Detection (FSAD)** aims to enable detecting anomalous samples with only a few normal samples as training data, and is an emerging topic in anomaly detection. We first distinguish FSAD from the semi-supervised anomaly detection setting (Ruff et al., 2019) where a limited number of labeled anomalies are available for training as it is sometimes referred to as few-shot anomaly detection in the literature (Pang et al., 2021a). The pioneering FSAD method of Sheynin et al. (2021) employs a hierarchical generative model to generate new samples from the few-shot examples. A discriminator is designed to discriminate generated images from the real ones and different transformations. Anomalies are then determined by whether the discriminator can correctly classify the type of transformations. Inspired by the few-shot learning paradigm, the RegAD method of Huang et al. (2022) uses a registration based proxy task for representation learning; this task aims to find the affine transformation that aligns the feature map of two samples from the same semantic class. RegAD requires additional related training data, for instance, data from other classes besides the target class on the MVTec dataset, for training the proxy task. The work of Ando & Yamamoto (2022) addresses a different few-shot setting that requires normal samples to be provided with semantic labels. When normal data comprises multiple semantic classes, embedding all normal samples into a single cluster may result in the failure to detect anomalies occuring between semantic classes. Learning multiple prototypes was proposed to tackle this issue. In comparion, our method adapts pretrained weights to target data using only a few normal training samples; unlike some of these other works, no additional data is required during the representation learning phase. This enables our method to be applied in a broader set of industrial anomaly detection scenarios.

**Contrastive Learning:** Pretraining feature representation through contrasting augmented samples of the same identity has demonstrated promising results. SimCLR (Chen et al., 2020; He et al., 2020) employed an N-pair loss (Sohn, 2016) to encourage two augmentations of the same instance (positive pair) to be close in the feature space and other instances (negative pair) to stay faraway. The existence of negative pairs requires large batchsize for training, BYOL (Grill et al., 2020) introduced an exponential moving average model to avoid collapsed predictions and get rid of negative pairs. Apart from representation pretraining, contrastive learning has been recently demonstrated to be effective for label efficient finetuning (Liu et al., 2021; Xu et al., 2022; Chen et al., 2022; Li et al., 2021b). When source and target domain data distributions are subject to covariance shift, contrastive training on the target data in a unsupervised fashion can potentially alleviate the domain shift (Xu et al., 2022; Li et al., 2021b). In this work, we demonstrate that contrastive training on the target domain data plays an important role in learning a good representation for downstream anomaly detection.

## 3 METHODOLOGY

In this work, we assume a model pretrained on large external image collection (e.g. ImageNet) is available. We refer to this external data as the source domain. Anomaly detection on industrial data is the task to be solved and is referred to as the target domain. We first describe contrastive training for adapting a pretrained model to the target domain distribution. We then introduce the cross-instance positive pair loss to encourage normal samples to form a cluster in the feature space. When prior knowledge on how to synthesize negative samples is available, we can introduce negative pairs to encourage better separation of normal and abnormal samples in the feature space. An overview of the proposed contrastive adaptation framework is shown in Fig. 1. Lastly, we describe how to build the density-based anomaly detection model on the learnt representations.

### 3.1 CONTRASTIVE TRAINING FOR ADAPTATION

We first denote the few-shot training examples from target domain as $\mathcal{D}_T = \{\mathbf{X}_i\}_{i=1\cdots N_T}$. The parameters of a backbone network are denoted as $\boldsymbol{\Theta}$ and $\mathbf{z} = f(\mathbf{X}; \boldsymbol{\Theta})$ encodes the input $\mathbf{X}$ into feature space. Contrastive training updates the model parameters $\boldsymbol{\Theta}$ by optimizing a contrastive loss in an unsupervised manner as in Eq. 1. In this work, we consider the BYOL (Grill et al., 2020) method for contrastive learning due to its smaller memory requirements.

$$\mathcal{L}_{Con} = -\frac{1}{N_T} \sum_{\mathbf{X}_i \in \mathcal{D}_T} \frac{q(g(\mathbf{z}_i))^\top g(\hat{\mathbf{z}}_i)}{||q(g(\mathbf{z}_i))|| \cdot ||g(\hat{\mathbf{z}}_i)||} \tag{1}$$

To learn effective representations, contrastive training contrasts between two random augmentations of the same input image, denoted as $t(\mathbf{X})$. The encoder network outputs the representation embedding for each augmented input as $\mathbf{z} = f(t(\mathbf{X}); \Theta)$. The representations are further projected to

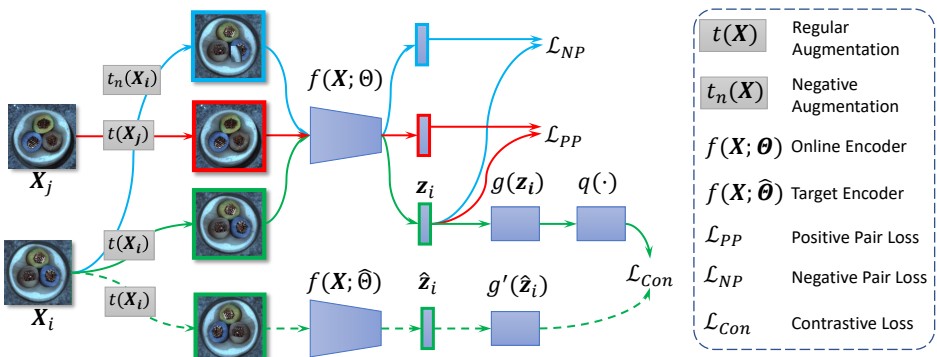

Figure 1: Illustration of adapting source domain pretrained model through combining contrastive training loss (green arrow lines), cross-instance positive pair loss (red arrow lines) and negative pair loss (blue arrow lines) for few-shot anomaly detection. The dashed arrow lines indicate no gradient backpropagation.

lower dimension through a projection head $g(\cdot)$. The cosine similarity is then calculated between the predictor's output $q(g(\mathbf{z}))$ on the online view and the projector's output $g(\hat{\mathbf{z}})$ on the target view. To avoid trivial solution, e.g. an encoder function giving constant outputs, the target view is the output of an exponential moving average model, i.e. $\hat{\mathbf{z}} = f(\mathbf{X}; \hat{\Theta})$ and $\hat{\Theta}_t = \beta\hat{\Theta}_{t-1} + (1 - \beta)\Theta_t$ where $\beta$ is a moving average hyperparameter. When a source domain model $\Theta^S$ is available, contrastive training on the target domain is initialized by the source domain model, i.e. $\Theta_0 = \Theta^S$, such that the low-level feature extractors can be reused. Therefore, contrastive training serves as adapting a pretrained network weights to the few-shot target domain training samples. A discussion on why contrastive training helps can be found in the Appendix A.1.

### 3.2 Cross-Instance Positive Pair Loss

The contrastive training objective encourages adaptation to target distribution, but this does not guarantee the learned representation is suitable for downstream density-based anomaly detection. Since anomaly detection inference is often implemented as fitting a multi-variate Gaussian distribution on the normal samples in the feature space, normal samples should ideally be embedded close to each other. Inspired by the success of one-class classification (Ruff et al., 2018) we propose to encourage normal samples to form a tight cluster in the feature space. Specifically, we treat a pair of randomly selected normal samples as a positive pair, the representations of each positive pair are encouraged to be closer by minimizing the cosine similarity as in Eq. 2 where $\mathbf{p}$ is a random permutation of the list $\{1, \cdots N\}$.

$$\mathcal{L}_{PP} = -\frac{1}{2N_T} \sum_i \sum_{j \in \mathbf{p}} \frac{f(t(\mathbf{X}_i);\Theta)^\top f(t(\mathbf{X}_j);\hat{\Theta})}{||f(t(\mathbf{X}_i);\Theta)|| \cdot ||f(t(\mathbf{X}_j);\hat{\Theta})||} + \frac{f(t(\mathbf{X}_j);\Theta)^\top f(t(\mathbf{X}_i);\hat{\Theta})}{||f(t(\mathbf{X}_j);\Theta)|| \cdot ||f(t(\mathbf{X}_i);\hat{\Theta})||} \quad (2)$$

Compared with the alternative of maintaining a fixed cluster center as proposed in (Ruff et al., 2018), the cross-instance positive pair loss has two advantages. First, we do not need to fix the cluster center at the start of training. This avoids introducing too much regularization on the representation embedding as the cluster center may vary during the course of training. Second, we minimize the cosine similarity between the online view and target view where the latter does not backpropagate gradients. This avoids collapse to a trivial solution (e.g. all zero weights) (Ruff et al., 2018). We note that the cross-instance positive pair loss is calculated on the features directly from the backbone network. This is due to the fact that backbone output feature will be used for anomaly detection so the loss should be optimized in the feature space.

### 3.3 Incorporating Negative Pair Loss

Synthesizing negative examples have been demonstrated to be successful in pretraining representation for anomaly detection. Well-calibrated synthesis approaches even achieved the state-of-the-art performance on certain datasets where the synthesized ones can match the real anomalies very

well (Li et al., 2021a). In this work, we propose to incorporate additional synthetic negative examples when prior knowledge is available. Specifically, we denote synthesizing negative sample as $t_n(\mathbf{X})$, to encourage better separation between normal and abnormal samples we minimize the cosine similarity between the original image embedding and the negative embedding as below:

$$\mathcal{L}_{NP} = \frac{1}{N_T} \sum_i \frac{f(t(\mathbf{X}_i); \hat{\Theta})^\top f(t_n(\mathbf{X}_i); \Theta)}{||f(t(\mathbf{X}_i); \hat{\Theta})|| \cdot ||f(t_n(\mathbf{X}_i); \Theta)||} \tag{3}$$

It is worth noting that the negative contrasting is also carried out directly on the backbone output features to reflect the constraints are applied to the feature representations. A relevant design was presented in (Ruff et al., 2019) for semi-supervised anomaly detection by minimizing the reciprocal of the distance between annotated anomalies and normal sample cluster center. Again, we believe minimizing the cosine similarity is compatible with the contrastive training objective and cross-instance positive pair loss with no risk of having a trivial solution. The final training loss combines the above three loss terms as $\mathcal{L}_{all} = \mathcal{L}_{Con} + \lambda_{PP}\mathcal{L}_{PP} + \lambda_{NP}\mathcal{L}_{NP}$.

### 3.4 DENSITY-BASED ANOMALY DETECTION

To perform anomaly detection using the learnt representations, we follow the density-based approach in (Li et al., 2021a) and fit a multivariate Gaussian distribution to the few-shot normal samples. Note that the learnt feature representations must be L2-normalized before density estimation and inference because during representation learning, we optimize the cosine similarity which is agnostic to the magnitude of feature representations. Moreover, to increase the amount of data for fitting the Gaussian distribution we produce $N_A$ times augmented samples from the few-shot normal samples. Formally, the mean $\mu$ and covariance $\Sigma$ is obtained through maximum likelihood estimation as below where $\mathcal{D}_{TA} = \underbrace{\mathcal{D}_T \cup \mathcal{D}_T \cup \cdots \mathcal{D}_T}_{N_A \text{ times}}$.

$$\mu = \frac{1}{|\mathcal{D}_{TA}|} \sum_{\mathbf{X}_i \in \mathcal{D}_{TA}} \frac{f(t(\mathbf{X}_i))}{||f(t(\mathbf{X}_i))||}, \quad \mathbf{\Sigma} = \frac{1}{|\mathcal{D}_{TA}|} \sum_{\mathbf{X}_i \in \mathcal{D}_{TA}} \left( \frac{f(t(\mathbf{X}_i))}{||f(t(\mathbf{X}_i))||} - \mu \right) \left( \frac{f(t(\mathbf{X}_i))}{||f(t(\mathbf{X}_i))||} - \mu \right)^\top \tag{4}$$

The anomaly score is then given by the Mahalanobis distance as in Eq. 5 and test samples are ranked by the anomaly score for anomaly detection.

$$d_{AS}(\mathbf{X}) = \sqrt{(f(\mathbf{X})/||f(\mathbf{X})||)^\top \mathbf{\Sigma}^{-1} (f(\mathbf{X})/||f(\mathbf{X})||)} \tag{5}$$

## 4 EXPERIMENTS

We evaluate the performance of our method on four industrial defect identification datasets and two datasets with synthetic common corruptions. We benchmarked against state-of-the-art anomaly detection methods and achieved very competitive performance. Finally, we carry out ablation studies on individual components and provide further insights into the negative pair loss.

### 4.1 DATASETS

We provide an overview of the datasets used in the experiments. **MVTec Dataset (Bergmann et al., 2019b)** contains 15 object categories, including 10 non-texture object categories and 5 texture object categories. Each category contains 60-300 normal samples for training and 30-400 normal and defect samples for testing. We follow the few-shot settings from (Sheynin et al., 2021) to create 2/5/10-shot anomaly detection protocols. **AITEX Dataset (Silvestre-Blanes et al., 2019)** is dedicated to detecting defects in textile fabric. This dataset consists of 140 normal sample images and 105 defect sample images with corresponding defect mask for localization. The original image resolution is 4096×256 pixels and the defects occupy a very small percentage of pixels. To allow for easier defect identification and mimicking a realistic defect identification procedure, we randomly crop out 5 patches of size $256 \times 256$ pixels from each image. Then 5/10/50-shot normal patches are randomly sampled as training examples. In total, test set consists of 600 normal patches and 105 defect patches. **Magnetic Tile Defects Dataset (Huang et al., 2020)** consists of 6 different types of defect magnetic tile surface images, namely "Blowhole", "Crack", "Fray", "Break", "Uneven", and "Free" (no defects). There are 1344 images in total of which 952 images contain defects. We

randomly select 5/10/50-shot normal examples from the "Free" class as training data and evaluate on the remaining images for anomaly detection. **SemiCon Dataset (Pahwa et al., 2021)** features 3D images collected from 3D X-ray microscopy (XRM) scans on integrated circuit packaging interconnects. The original dataset contains 53 3D images of memory dies and the anomaly detection task is to identify voids in the solder region: a large void to solder ratio may indicate defective products. To create an anomaly detection benchmark, we slice the 3D images to obtain 2D images. 5/10/50-shot normal images are randomly selected for training, 110 defect samples and 500 normal samples are used for testing. **CIFAR10/100-C (Hendrycks & Dietterich, 2019)** are adopted to simulate industrial defects through synthesizing common corruptions. We propose an anomaly detection protocol by treating all corrupted test samples as anomalies for each class. The dataset contains 5 levels of 15 different corruptions, e.g., noises, blurry, foggy. For easy comparison we choose 9 types of level 2 corruptions. 5000 clean testing samples from each class of original CIFAR10/100 dataset are used as normal testing samples, the corresponding corrupted samples of that class are treated as defect testing samples. We evaluate at 10-shot for both datasets. A few examples of anomalies for the industrial datasets are presented in Fig. 2 with more examples to be found in the Appendix A.4.

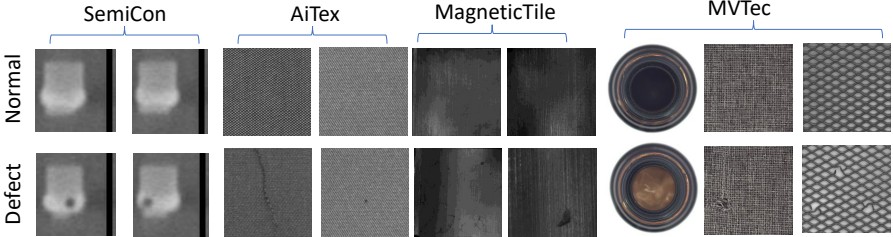

Figure 2: Examples of industrial image data used for anomaly detection.

## 4.2 COMPETING METHODS

We compare against multiple anomaly detection methods in the experiments. We first benchmark the vanilla auto-encoder (**AE**) employed in (Bergmann et al., 2019a) which is trained to reconstruct input images and the difference between input and reconstructs measures the anomaly score. **VAE** (Kingma & Welling, 2014) imposes constraints on the latent variables to be Gaussian. Multiple samples are drawn in the latent space and decoded to image space for measuring anomaly score (An & Cho, 2015). **DeepSVDD** (Ruff et al., 2018) trains the network by forcing normal training samples to embed close to a cluster center. The learned cluster center can later serve as the prototype for anomaly detection by measuring the distance as anomaly score. **CutPaste** (Li et al., 2021a) introduced CutPaste as an augmentation approach to synthesize negative examples for pretraining feature representation network. It is worth noticing that CutPaste mimics the real anomalies that would appear in the MVTec dataset, therefore the effectiveness of CutPaste may not generalize to other types of anomalies. **RotNet** (Golan & El-Yaniv, 2018) proposed a self-supervised pretraining approach by predicting the augmentations (rotations) applied. This approach is most effective in anomaly detection on natural semantic images and the advantage may disappear on industrial images where the pose of background is naturally more diverse. **CSI (Tack et al., 2020) proposed to contrast distribution shifted augmented images with original images to increase the gap between normal and abnormal samples. This is similar to maintaining only the negative pair loss proposed in this work. CFLOW-AD (Gudovskiy et al., 2022) adopted a conditional normalizing flow model for fast anomaly localization. We adapt CFLOW-AD to training on few-shot anomaly detection task.** **TDG** (Sheynin et al., 2021) proposed to employ generative model for few-shot anomaly detection by differentiating image patches into either fake or one of a list of predefined augmentations. **DifferNet** (Rudolph et al., 2021) estimates density through normalizing flow with a few supporting training samples. The above two approaches are compared on the MVTec dataset. **Ours (w/o np)** optimizes on the combination of contrastive loss and cross-instance positive pair loss. Assuming prior knowledge on the anomaly is available, **Ours (w/ np)** incorporates the negative pair loss and optimizes on the combination of all three losses. Among these methods, CutPaste and Ours (w/ np) are built upon the prior knowledge of anomalies while other methods do not make explicit assumptions on how the anomalies would look like.

## 4.3 EXPERIMENT DETAILS

**Training Details**: For all experiments, we use the ResNet18 (He et al., 2016) backbone for feature extraction. For all competing methods, we initialize backbone weights with ImageNet pretrained

weights. We set the weight for cross-instance positive pair loss as 0.8 and the weight for negative pair loss as 0.6. We use the Adam optimizer Kingma & Ba (2015) for all experiments with learning rate initialized to $3 \times 10^{-4}$, $\beta_1 = 0.9$ and $\beta_2 = 0.99$. We fix the batch size to 64, thus creating 64 pairs for contrastive training. To generate cross-instance pairs, we randomly permute the 64 images and each of the 64 image is paired with a randomly permuted one, resulting in 64 positive pairs. Similarly, we pair each image to its negatively augmented one to create another 64 negative pairs. For density model fitting, we use $N_A = 10$. The area under the ROC (AUROC) is used to assess performance, where anomalies are treated as the positive class.

**Data Augmentation**: For the negative pair loss, we synthesize negative examples $t_n(\mathbf{X})$ with Cut-Paste augmentations (Li et al., 2021a) for MVTec datasets. For all industrial datasets, regular data augmentation $t(\mathbf{X})$ comprises affine transformation and color manipulations (e.g. blurring and grayscaling). For CIFAR10/100-C, regular augmentation only includes affine transformation and negative augmentation comprises blurring and randomly perturbing image brightness and contrast.

## 4.4 FEW-SHOT ANOMALY DETECTION ON INDUSTRIAL DATASETS

In this section, we explore identifying defects on industrial images. We first evaluate the few-shot anomaly detection performance on **MVTec** dataset with results in Tab. 1. We make the following observations. First, without any specific prior knowledge on the anomalies, our method (Ours (w/o np)) outperforms all competing methods by a clear margin. Furthermore, with prior knowledge on the potential anomalies, our method (Ours (w/ np)) still outperforms CutPaste with the same negative augmentations in the 2-shot and 5-shot settings. We are only slightly behind CutPaste in the 10-shot case. Both observations suggest the effectiveness of contrastive adaptation and cross-instance positive pair loss. We further observe that CutPaste exhibits a significant lead on leather, wood and toothbrush images. We attribute this to the fact that these categories contain many anomalies that can be synthesized from CutPaste and scar augmentation: the "cut" defect for the "leather" category, the "scratch" defect for "wood" and scar like defects for "toothbrush". In contrast, our method relies more on adaptation from the pretrained model and learning from few-shot normal samples. As a result, its performance is generally better on more diverse types of objects: in the 2-shot setting, our method outperforms CutPaste on 8 of the 15 categories while CutPaste is winning on 4/15 categories.

Table 1: Few-shot anomaly detection on MVTec dataset. Per-category AUROC is reported for all competing methods. All numbers are in %. The results of DiffNet* and TDG* are derived from (Sheynin et al., 2021), where − indicates per-class results are not available.

| | | bottle | cable | caps. | hazel. | metal. | pill | screw | tooth. | transis. | zipper | carpet | grid | leather | tile | wood | avg. |
|---|---|---|---|---|---|---|---|---|---|---|---|---|---|---|---|---|---|
| 2 shot | AE | 73.49 | 64.22 | **62.43** | 73.54 | 35.97 | 75.19 | 35.56 | 73.33 | 48.92 | 40.73 | 22.11 | 45.13 | 31.52 | 73.35 | 58.42 | 53.26 |
| | VAE | 68.73 | 61.83 | 60.47 | **73.82** | 41.20 | 76.08 | 39.98 | 72.22 | 68.83 | 38.16 | 25.08 | 40.85 | 37.40 | 72.69 | 44.91 | 54.75 |
| | DeepSVDD | 85.79 | **66.06** | 51.62 | 53.39 | 50.44 | **77.27** | 51.23 | 69.72 | 58.04 | 59.30 | 70.06 | 38.10 | 40.42 | **81.02** | 51.23 | 60.28 |
| | Ours (w/o np) | **88.06** | 59.05 | 58.82 | 60.77 | **68.51** | 65.55 | 56.48 | 73.63 | 72.84 | 70.74 | 76.00 | 61.18 | 63.11 | 68.25 | **77.85** | **68.06** |
| | CutPaste | 86.39 | 64.59 | **61.56** | 73.59 | 49.92 | 66.91 | 41.93 | **82.04** | 55.53 | 59.97 | 52.26 | 46.25 | **83.82** | 71.28 | 84.79 | 65.38 |
| | Ours (w/ np) | **90.95** | 66.55 | 59.38 | 68.82 | **68.96** | 66.68 | 54.90 | 74.81 | 74.33 | 72.82 | 73.18 | 61.77 | 65.28 | 69.64 | 79.85 | **69.86** |
| 5 shot | AE | 76.59 | 65.59 | 72.92 | 73.64 | 49.61 | 76.73 | 40.32 | 75.00 | 64.79 | 59.95 | 38.76 | 42.61 | 43.72 | 66.56 | 74.04 | 61.39 |
| | VAE | 70.24 | 62.52 | 73.08 | 74.57 | 41.96 | 76.55 | 41.55 | 79.51 | 72.92 | 59.50 | 39.32 | 56.16 | 45.99 | 60.22 | 76.03 | 63.05 |
| | DeepSVDD | 86.19 | 68.29 | 60.23 | 54.07 | 52.16 | 78.42 | 52.15 | 81.39 | 69.04 | 74.80 | 51.54 | 51.55 | 58.20 | 82.83 | **93.51** | 67.59 |
| | CSI | 80.55 | 60.48 | 62.07 | 74.40 | 59.83 | 69.29 | 32.71 | 79.44 | 55.71 | 63.71 | 56.17 | 39.53 | 51.56 | 55.38 | 75.96 | 61.12 |
| | CFLOW-AD | **98.17** | **81.65** | 73.12 | 88.25 | 74.88 | 68.22 | 45.09 | 82.50 | **84.79** | 83.53 | 73.48 | 50.96 | **87.84** | **91.59** | 92.37 | **78.43** |
| | DifferNet* | - | - | - | - | - | - | - | - | - | - | - | - | - | - | - | 72.10 |
| | TDG* | - | - | - | - | - | - | - | - | - | - | - | - | - | - | - | 77.90 |
| | Ours (w/o np) | 94.58 | 70.67 | **74.06** | 78.29 | 79.66 | 81.56 | 62.70 | 87.18 | 78.08 | 75.03 | 84.67 | 62.31 | 74.90 | 78.34 | 89.66 | 78.11 |
| | CutPaste | **98.41** | **80.32** | 69.20 | **89.90** | 72.24 | **82.85** | 59.13 | **90.89** | 68.56 | 68.89 | 73.13 | 49.94 | **83.93** | **91.50** | **96.08** | 78.33 |
| | Ours (w/ np) | 95.73 | 78.05 | **74.26** | 86.89 | **78.57** | 81.31 | **62.26** | 88.81 | 74.80 | **75.70** | 79.90 | **61.85** | 77.92 | 75.34 | 90.12 | **78.76** |
| 10 shot | AE | 81.91 | 69.34 | 73.54 | 74.14 | 57.72 | 78.40 | 50.07 | 93.11 | 66.49 | 60.29 | 41.43 | 49.96 | 45.07 | 72.62 | **95.70** | 64.92 |
| | VAE | 82.06 | 64.05 | 73.59 | 75.04 | 56.74 | 78.07 | 50.03 | 91.66 | 73.41 | 60.19 | 44.78 | 56.05 | 47.45 | 76.55 | 94.56 | 67.02 |
| | DeepSVDD | 86.75 | 68.85 | 65.05 | 74.04 | 70.82 | 78.42 | 53.13 | 86.39 | 69.08 | 77.28 | 52.07 | 51.71 | 58.27 | 86.33 | 94.02 | 71.48 |
| | CSI | 83.44 | 62.65 | 64.30 | 77.07 | 61.98 | 71.78 | 33.89 | 82.30 | 57.71 | 66.00 | 58.19 | 40.95 | 53.41 | 57.37 | 78.69 | 63.32 |
| | CFLOW-AD | **99.50** | **84.58** | 75.75 | **91.42** | 77.57 | 70.67 | 46.71 | 85.47 | **87.84** | 86.53 | 76.12 | 52.79 | **91.00** | 94.88 | 95.69 | **81.10** |
| | DifferNet* | - | - | - | - | - | - | - | - | - | - | - | - | - | - | - | 73.60 |
| | TDG* | - | - | - | - | - | - | - | - | - | - | - | - | - | - | - | 78.00 |
| | Ours (w/o np) | 97.84 | 71.07 | **79.23** | 78.72 | **80.57** | 82.81 | 61.85 | 95.12 | 87.21 | 83.71 | **84.73** | 63.10 | 76.88 | 81.14 | 89.69 | 80.92 |
| | CutPaste | 98.71 | 81.83 | **83.15** | **94.47** | **88.92** | 85.63 | **64.55** | 91.50 | 70.01 | **86.90** | **83.92** | 55.13 | **99.50** | **91.81** | 96.21 | **84.82** |
| | Ours (w/ np) | **99.03** | **83.49** | 78.38 | 87.25 | 79.47 | 82.55 | 63.92 | **94.71** | 87.21 | 84.45 | 79.95 | **63.14** | 78.88 | 81.14 | **97.15** | 82.71 |

We further evaluate defect identification performance on another three industrial datasets, namely **SemiCon**, **AITEX** and **MagneticTile**, with results in Tab. 2. The following observations are drawn from the results. First, without any prior knowledge, our model achieves the state-of-the-art performance under lower budgets of available training samples (5 and 10 shots) on all three datasets. It is only slightly behind DeepSVDD at 50-shot on AITEX. This suggests adapting pretrained models to

target domain is effective for realistic industrial image defect identification tasks. Second, methods demonstrating strong performance on MVTec dataset may not generalize to other types of defects. For example, while CutPaste is one of the best performing methods on MVTec, its performance on SemiCon and AITEX is much worse than more traditional approaches. One potential reason for this poor performance is that the synthesized negative samples used in CutPaste are not representative of the defects in SemiCon and AITEX datasets; a more detailed analysis can be found in the Appendix.

Table 2: Few-shot defect identification results on additional three industry image datasets. AUROC is reported as evaluation metrics. All numbers are in (%).

|  | SemiCon | | | AITEX | | | Magnetic Tile | | |
|---|---|---|---|---|---|---|---|---|---|
|  | 5-shot | 10-shot | 50-shot | 5-shot | 10-shot | 50-shot | 5-shot | 10-shot | 50-shot |
| AE | 65.38 | 70.24 | 71.74 | 47.01 | 60.59 | 63.30 | 51.58 | 52.66 | 54.70 |
| VAE | 72.42 | 73.53 | 80.14 | 52.99 | 66.36 | 67.88 | 51.90 | 53.40 | 54.30 |
| DeepSVDD | 52.02 | 71.40 | 79.09 | 70.15 | 74.40 | **80.07** | 54.80 | 55.90 | 57.73 |
| RotNet | 55.13 | 75.55 | 80.45 | 71.20 | 75.19 | 80.02 | 55.69 | 57.14 | 60.00 |
| CutPaste | 48.71 | 71.60 | 75.03 | 50.23 | 69.58 | 79.83 | 57.82 | 61.00 | 62.14 |
| Ours (w/o np) | **78.87** | **80.72** | **82.00** | **73.44** | **77.45** | 78.14 | **58.23** | **61.93** | **63.55** |

## 4.5 DETECTING SYNTHETIC NOISE CORRUPTIONS

In this section, we evaluate on detecting synthetic noise corruptions as anomalies. The synthesized corruptions mimic noise patterns that are commonly seen in industrial environments. We benchmark on **CIFAR10-C** and **CIFAR100-C** with 10-shot training samples for this purpose. In a similar fashion to MVTec, we adapt ImageNet pretrained weights to each individual semantic class. For CIFAR100-C, we choose the 20 superclasses as the semantic class for simplicity. We present the anomaly detection results on each semantic category of CIFAR10-C in Tab. 3. We first observe from the results that in average our methods, both with and without prior knowledge, lead the competing methods with a clear margin. The closest competing method, CutPaste, is 4% lower than Ours (w/ np). This is in contrast to the extraordinary performance demonstrated on MVTec by CutPaste. We attribute this to the fact that the corruptions in CIFAR10-C are diverse and may not be easily synthesized by the augmentation methods specifically tailored for the MVTec dataset. We further benchmark on CIFAR100-C and compare with DeepSVDD and CutPaste. We draw similar conclusions from the results. Our method is still stronger than CutPaste with a clear margin. This is caused by the mismatch between the negative samples synthesized by CutPaste and corruptions in the dataset.

Table 3: 10-shot anomaly detection on CIFAR10-C dataset. All numbers are reported as AUROC in %.

|  | Airpl. | Auto. | Bird | Cat | Deer | Dog | Frog | Horse | Ship | Truck | Avg. |
|---|---|---|---|---|---|---|---|---|---|---|---|
| AE | 54.38 | 53.67 | 54.86 | 53.79 | 56.15 | 54.3 | 51.70 | 54.79 | 55.89 | 50.38 | 53.99 |
| VAE | 57.47 | 52.17 | 56.32 | 56.11 | 58.67 | 57.09 | 59.18 | 54.06 | 56.90 | 50.61 | 55.86 |
| DeepSVDD | 63.27 | 61.64 | 60.75 | 60.71 | 61.54 | 62.11 | 63.00 | 59.83 | 63.72 | 67.29 | 62.37 |
| RotNet | 69.21 | 53.21 | 66.03 | 64.00 | 67.30 | 67.84 | 70.20 | 70.84 | 59.05 | 63.04 | 64.07 |
| CutPaste | 58.62 | 73.01 | 70.56 | 75.57 | 68.65 | 71.32 | 72.26 | 70.82 | **72.53** | 66.79 | 70.01 |
| Ours (w/o np) | 71.68 | 64.68 | 75.74 | 74.38 | 76.55 | 74.69 | **81.40** | **79.10** | 60.02 | 74.53 | 72.27 |
| Ours (w/ np) | 72.23 | 67.66 | 78.41 | 77.45 | 75.33 | 75.20 | 75.35 | 75.88 | 72.20 | 75.85 | 74.56 |

Table 4: Anomaly detection on CIFAR100-C dataset. Per super class performance is reported. All numbers are reported as AUROC in %.

|  | Sup.Cls. 0 | Sup.Cls. 1 | Sup.Cls. 2 | Sup.Cls. 3 | Sup.Cls. 4 | Sup.Cls. 5 | Sup.Cls. 6 | Sup.Cls. 7 | Sup.Cls. 8 | Sup.Cls. 9 |
|---|---|---|---|---|---|---|---|---|---|---|
| Deep SVDD | 75.10 | 73.48 | 70.35 | 63.52 | 73.53 | 76.19 | 69.91 | 74.76 | **79.70** | 67.13 |
| CutPaste | 75.96 | **69.59** | 70.51 | 75.71 | 72.50 | 74.14 | **76.22** | 76.69 | 71.79 | 78.31 |
| Ours (w/ np) | **77.61** | 66.69 | **77.84** | **76.21** | **79.93** | **78.17** | 73.33 | **78.80** | 74.63 | **81.17** |

|  | Sup.Cls. 10 | Sup.Cls. 11 | Sup.Cls. 12 | Sup.Cls. 13 | Sup.Cls. 14 | Sup.Cls. 15 | Sup.Cls. 16 | Sup.Cls. 17 | Sup.Cls. 18 | Sup.Cls. 19 | Avg |
|---|---|---|---|---|---|---|---|---|---|---|---|
|  | 70.00 | 73.17 | 60.82 | 56.91 | 66.36 | 65.81 | 60.60 | 69.88 | 70.52 | 71.30 | 69.24 |
|  | 80.28 | **75.83** | 77.92 | 78.56 | 70.87 | 78.84 | 72.83 | 71.63 | 72.97 | 75.61 | 74.84 |
|  | **81.27** | 75.59 | **80.11** | **79.98** | 75.56 | **80.04** | **73.65** | **77.54** | **77.16** | **78.13** | **77.17** |

## 4.6 ABLATION STUDY

In this section, we investigate the effectiveness of the individual components using the CIFAR10-C dataset. In particular, we demonstrate the importance of contrastive training on few-shot target domain normal samples (Contrast. Train), incorporating cross-instance positive pair loss (Positive Pair Loss) and incorporating the negative pair loss (Negative Pair Loss). We further evaluate incorporating L2 normalization (L2 Norm) during anomaly detection density model fitting and inference.

We present the ablation study results in Tab. 5. We make the following observations from the results. First, as we expected, reusing ImageNet pretrained weights for downstream anomaly detection yields significant improvement in performance ($52.54\% \rightarrow 67.32\%$). This suggests the significance of a good representation for anomaly detection. Adapting pretrained model to the target distribution through contrastive training further improves $2\%$ in average ($67.32\% \rightarrow 70.05\%$). To encourage feature embedding suitable for density-based anomaly detection, we further incorporate the positive pair loss and this again yields additional $3\%$ improvement ($70.05\% \rightarrow 73.68\%$). As an alternative approach, one could encourage all normal samples' features to embed close to a fixed cluster center (F.C.) following Ruff et al. (2018). However, as the cluster center must be fixed through the first forward pass, this could pose too much constraint on the representation learning and yield inferior results ($68.21\%$). Finally, when we combine negative pair loss, this gives a final boost of performance to $74.56\%$. We also hypothesize that L2 normalization on feature representation is necessary and the ablation study validated the hypothesis. By removing the L2 normalization on anomaly inference features the performance drops from $74.56\%$ to $72.11\%$, indicating the normalization is essential to fitting better density model and distance-based anomaly detection.

Table 5: Ablation study on CIFAR10-C 10-shot FSAD. CIPP stands for cross-instance positive pair.

| Pretrained Weights | Contrast. Train | Positive Pair Loss | Negative Pair Loss | L2 Norm | Avg. Acc. |
|---|---|---|---|---|---|
| - | - | - | - | ✓ | 52.54 |
| ImageNet | - | - | - | ✓ | 67.32 |
| ImageNet | ✓ | - | - | ✓ | 70.05 |
| ImageNet | ✓ | F.C. (Ruff et al., 2018) | - | ✓ | 68.21 |
| ImageNet | ✓ | C.I.P.P. | - | ✓ | 73.68 |
| ImageNet | ✓ | C.I.P.P. | ✓ | ✓ | **74.56** |
| ImageNet | ✓ | C.I.P.P. | ✓ | - | 72.11 |

### 4.6.1 WHEN DOES INCORPORATING NEGATIVE EXAMPLES HELP?

As we discussed, incorporating negative example during adaptation is not always beneficial. The advantage hinges on whether prior knowledge on abnormal sample distribution is available. To verify this point, we evaluate incorporating negative pair loss on 3 industry datasets, SemiCon, AITEX and Magnetic Tile. SemiCon dataset features anomalies that are quite different from MVTec, while AITEX and Magnetic Tile are relatively more similar to the texture categories of MVTec. We choose CutPaste (Li et al., 2021a) as negative augmentation for these 3 datasets. The results in Tab. 6 demonstrate that when the negative augmentation is substantially different from the real anomalies, e.g. the void circles in SemiCon dataset, incorporating negative pair loss with inappropriate augmentation will harm performance ($78.87 \rightarrow 62.97\%$). On the contrary, when anomalies can be simulated, even imperfectly, e.g. Magnetic Tile dataset, incorporating negative pair loss will further improve performance. Overall, we conclude that incorporating negative pair loss is only helpful when prior knowledge on the potential anomalies is concrete and anomalies can be simulated through negative augmentation. Our model without negative pair loss is suitable for tasks without prior knowledge or when generating negative augmentation is difficult.

Table 6: Effect of negative pair loss with CutPaste augmentation on industry image datasets.

| | SemiCon | AITEX | Magnetic Tile |
|---|---|---|---|
| **Ours (w/o np)** | 78.87 | 73.44 | 58.23 |
| **Ours (w/ np)** | 76.29 | 65.54 | 59.45 |

## 5 CONCLUSION

Industry defect inspection requires the capability of anomaly detection with very limited normal samples for training. To meet this demand, we proposed a few-shot anomaly detection approach through adapting models pretrained on large external image collections to few-shot normal samples from the target task. We achieve this adaptation by optimizing a contrastive loss and cross-instance positive pair loss. When prior knowledge on possible anomalies is available we further incorporate a negative pair loss to separate normal sample embeddings from the synthesized negative samples. We extensively evaluated the performance of the proposed method on four real industrial defect detection datasets and two synthetic datasets mimicking realistic corruptions. Our method achieved state-of-the-art performance on all datasets when only a handful of normal samples are available. Finally, we show that the benefit of using synthetic negative samples is task-dependent and should only be considered when accurate prior knowledge is available.

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

## A APPENDIX

### A.1 DISCUSSIONS ON CONTRASTIVE TRAINING HELPING ADAPTATION

Recent works have demonstrated that contrastive training helps adapt model parameters to target domain distributions (Liu et al., 2021; Xu et al., 2022; Chen et al., 2022; Li et al., 2021b). We argue that contrastive training on target domain data can alleviate the negative impact of covariate shift. For simplicity, we denote the source domain dataset as $\mathcal{D}_S$, e.g. the ImageNet dataset, and the target domain dataset as $\mathcal{D}_T$, e.g. anomaly detection dataset. The objective of supervised training on the source domain can be seen as minimizing the following cross entropy loss where $h(\cdot)$ is the classifier on source domain and $f(\cdot)$ is the backbone network to be transferred.

$$\Phi^{S*}, \Theta^{S*} = \arg\min_{\Phi,\Theta} \frac{1}{|\mathcal{D}_S|} \sum_{\mathbf{X}_i, y_i \in \mathcal{D}_S} \mathcal{L}_{CE}(h(f(\mathbf{X}_i; \Theta); \Phi), y_i) \qquad (6)$$

Covariate shift between source and target dataset indicates a distributional misalignment, i.e. $p_S(\mathbf{X}) \neq p_T(\mathbf{X})$ which is easily manifested by the difference in the contents of the source and target domain data. Therefore, it is reasonable to believe the backbone network optimized for source domain model is not optimal for the target domain distribution. To ease the negative impact caused by covariate shift, we introduce contrastive training on target domain by optimizing an unsupervised contrastive loss with model parameters initialized by the source domain ones, as in Eq. 7.

$$\Theta^{T*} = \arg\min_{\Theta} \frac{1}{|\mathcal{D}_T|} \sum_{\mathbf{X}_i \in \mathcal{D}_T} \mathcal{L}_{Con}(f(t(\mathbf{X}_i); \Theta), f(t(\mathbf{X}_i); \hat{\Theta})), \quad s.t. \; \Theta_0^T = \Theta^{S*} \qquad (7)$$

By minimizing the contrastive loss, the network is able to capture key features from the target domain to discriminate non-identical instances and we empirically demonstrate this to be effective for adapting source model to a target domain for downstream anomaly detection. We further provide another perspective into the effectiveness of contrastive learning. When the augmentations are chosen to be mimic the commonly seen variations within normal samples, contrasting two augmented images forces the network to produce similar representations regardless of the augmentations. This means the representation learned from contrastive training allows the network to learn features invariant to common variations in appearance and pose that one could encounter in industrial imaging environments. Such ability will help bring normal samples closer in the feature space, thus benefit downstream anomaly detection.

### A.2 FURTHER ANALYSIS

In this section, we provide additional evaluations on qualitative examination of representation learning and discuss the when incorporating negative samples should be employed.

### A.2.1 QUALITATIVE EXAMINATION OF REPRESENTATION LEARNING:

The benefit of incorporating each of the proposed three losses for adaptation to target anomaly detection are validated through empirical experiments on multiple datasets. In this section, we provide qualitative observations into the advantage of incorporating these losses through t-SNE visualization of test data representations Van der Maaten & Hinton (2008). Specifically, we randomly select 1,500 testing samples from CIFAR10-C dataset for visualization. The feature points are projected into 2D space and visualized in Fig. 3. The feature embedding with ImageNet pretrained weights only, (a) w/o Adaptation, shows a substantial overlap between normal and abnormal samples. When contrastive training is applied, (b) w/ Contrastive Loss, we observe a clear seperation between normal and abnormal samples. When additional positive pair loss, (c) PP Loss, and negative pair loss, (d) w/ NP Loss, are incorporated, the normal samples are further grouped into a tighter cluster with larger distrinction between normal and abnormal samples.

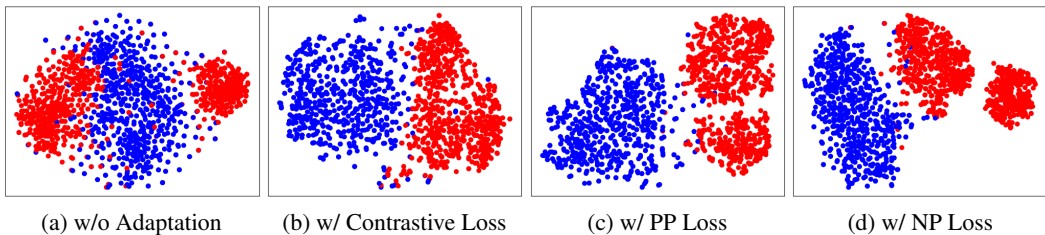

| (a) w/o Adaptation | (b) w/ Contrastive Loss | (c) w/ PP Loss | (d) w/ NP Loss |

Figure 3: T-SNE visualization of anomaly detection testing data on selected testing samples from CIFAR10-C dataset. Blue and red colors indicate normal and abnormal samples respectively.

### A.3 COMPARING REAL ANOMALIES V.S. NEGATIVE AUGMENTATION

In this section, we provide visual examples for both real anomalies and negative augmentation obtained through CutPaste Li et al. (2021a). We randomly sample 4 normal training samples from SemiCon dataset and augment the selected samples through CutPaste. The synthesized negative examples are compared with 4 randomly selected real anomalies in Fig. 4. The qualitative examination indicate that CutPaste augmentation may not be suitable for all types of industrial images and this is reflected in the relatively poor performance of CutPaste and Ours (w/ np) on SemiCon and AITEX dataset.

### A.4 ADDITIONAL VISUALIZATION OF INDUSTRIAL DATASETS

We provide more examples of the industrial datasets used in this work, namely the SemiCon, AITEX and Magnetic Tile. We illustrate both normal and defect samples of AITEX and Magnetic Tile in Fig. 5, SemiCon in Fig. 4. We notice that most defects in the SemiCon datasets are caused by the circular void in the solder area. The defects in AITEX are often curvilinear and the background is dense with repetitive patterns. The defects for Magnetic Tile is more subtle.

### A.5 DISTRIBUTION OF ANOMALY SCORES

In this section, we further compare different ablated models through visualizing the distribution of anomaly scores on CIFAR10-C few-shot anomaly detection task. Specifically, we compare a) w/o contrastive finetuning; b) w/ contrastive finetuning; c) additionally w/ PP loss; and d) additionally w/ NP loss. The results in Fig. 6. We make the following observations. First, directly using the model pretrained on ImageNet, a) w/o contrastive finetuning, failed to differentiate normal and abnormal samples as the two distributions are almost the same. With contrastive finetuning on target data, b) w/ contrastive finetuning, we see a clear gap between normal and abnormal distributions, suggesting better anomaly detection performance. By further incorporating the positive pair loss, c) w/PP loss, the gap between two distributions are more significant. Finally, incorporating negative pair loss, d) w/ NP loss, is most effectiv for differentiating normal from abnormal samples.

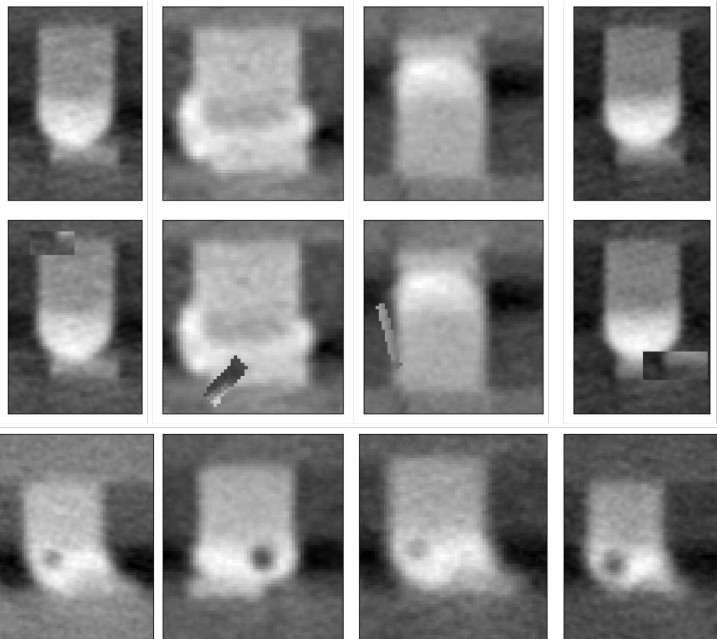

Figure 4: Samples of applying CutPaste on industry dataset SemiCon. First row is the original samples, and second row is the corresponding generated sample from CutPaste augmentation. Third row shows a few samples of the real defective images.

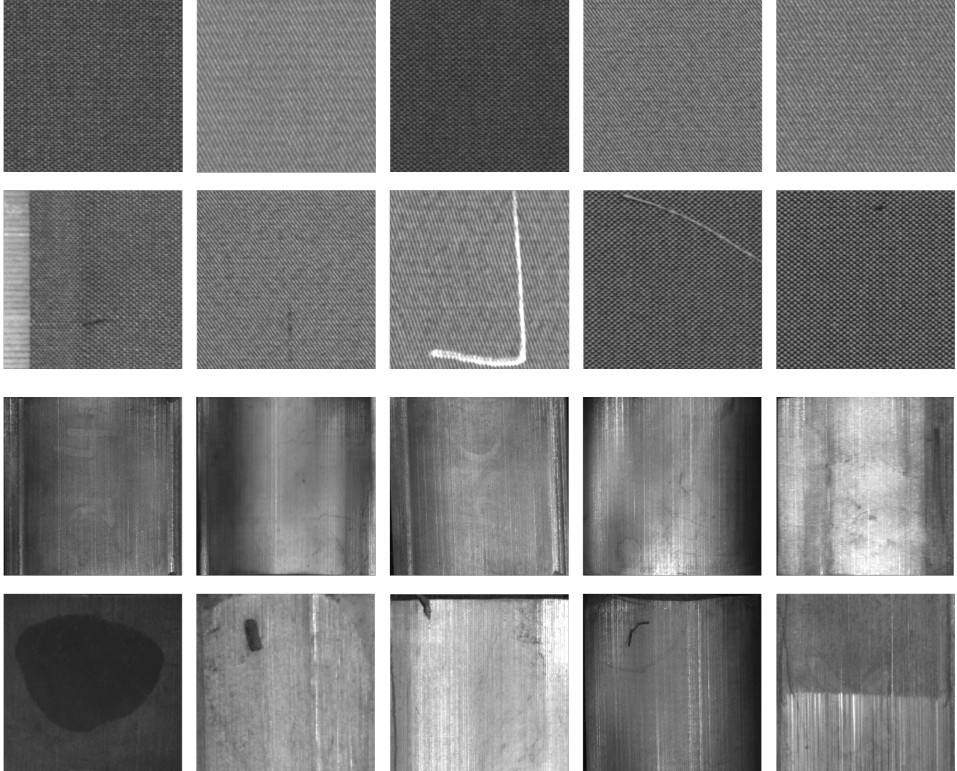

Figure 5: Illustration of Aitex(upper) and Magnetic Tile dataset. The first row of each dataset are normal examples, second rows are defective examples.

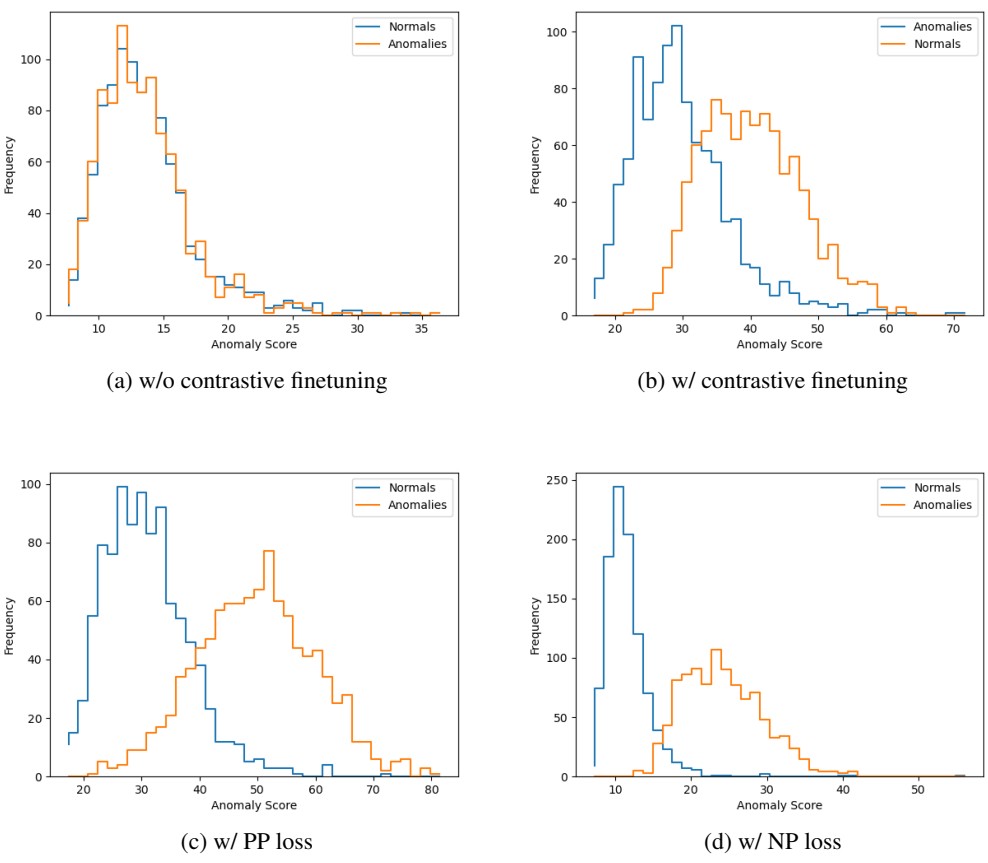

Figure 6: Comparing different ablated models through anomaly score distributions.

## A.6 EVALUATION ON ADDITIONAL BACKBONE

We evaluate the effectiveness of the proposed representation learning approach with stronger backbone network. In specific, we evaluate ResNet101 on the SemiCon dataset in a few-shot anomaly detection setting. The results in Tab. 7 reveal that contrastive finetuning and across instance positive pair are also effective with stronger backbone network.

Table 7: 5-shot anomaly detection with ResNet101 on SemiCon dataset.

| Contrast. Train | Positive Pair Loss | Avg. Acc. |
|:---:|:---:|:---:|
| - | - | 67.72 |
| ✓ | - | 84.22 |
| ✓ | ✓ | 86.65 |

## A.7 IMBALANCED ANOMALY DETECTION

We notice that the original anomaly detection dataset is already highly imbalanced. For example, the AiTex dataset consists of 600 normal and 100 abnormal samples in the testing set. Since the ratio between normal and abnormal samples does not affect model training, we further implemented a controlled experiment where we manually decrease the number of anomalies in the SemiCon dataset. Specifically, we fix the normal samples to 500 and randomly subsample 5, 10, 20 and 50 abnormal samples for evaluation. The results in Tab. 8 demonstrate the superiority of proposed method against existing competing methods.

Table 8: Generated by Spread-LaTeX

| #Anomalies | 5 | 10 | 20 | 50 |
|:---|:---:|:---:|:---:|:---:|
| deepsvdd | 52.01 | 49.71 | 49.97 | 52.40 |
| cutpaste | 86.31 | 83.82 | 72.82 | 74.31 |
| VAE | 80.56 | 78.12 | 67.67 | 73.91 |
| AE | 62.12 | 65.52 | 76.83 | 74.39 |
| Ours | 88.36 | 86.50 | 85.87 | 80.72 |

