# OpenReview forum: "Few-Shot Anomaly Detection on Industrial Images through Contrastive Fine-Tuning"
_ICLR.cc/2023/Conference — Submitted to ICLR 2023_

### Official Review · Reviewer_n46i · 2022-10-24

**Confidence:** 3
**Correctness:** 3
**Technical Novelty And Significance:** 2
**Empirical Novelty And Significance:** 2
**Recommendation:** 6

**Clarity, Quality, Novelty And Reproducibility:**

The paper is clear in general but could be better illustrated with visualization of multivariate Gaussian for additional interpretation and insights.

The novelty of this paper is limited in the sense that most key steps are very similar to the SOTA method CSI, although using domain adaptation for few-shot setting is somehow interesting and useful in most applications.

The implementations are well-elaborated and can be reproduced.


**Strength And Weaknesses:**

Strength:
* One of the major strengths of this paper is the extensive experiments on six benchmarks and convincing results.
* Another strength is the idea of adapting pre-trained networks to target domains through contrastive learning under few-shot AD.

Weakness:
* It would be helpful to provide visualization after the outputs fit the multivariate Gaussian. This may further demonstrate if positive/negative pairs are grouped/separated as expected.
* Some experiment details are missing, e.g., how many normal images were used in Sec. 4.5.
3* Some competitive methods such as TDG (as mentioned in the paper) could be added to the experiments on cifar10-c and cifar100-c.
* The state-of-the-art CSI [1] can run under few-shot setting and could be evaluated here, too. Note most key steps in this paper are very similar to CSI, so it is strongly recommended to compare with CSI in few-shot tests.
* It should be noted that in Fig. 1 after negative augmentation, the image however did not change to an anomaly, as expected.

[1] Jihoon, et al; CSI: Novelty Detection via Contrastive Learning on Distributionally Shifted Instances.

**Summary Of The Paper:**

This paper focuses on few-shot anomaly detection where only a few normal samples are available in training. The new method leverages contrastive learning to transfer pre-trained from the source domain to target domains supported by few-shot samples. In addition, the instance positive pair loss is used to tight up normal samples, while incorporating negative pair loss to separate the anomaly and normal samples (with prior and generated samples). Last, the multivariate Gaussian and density method are used for final detection.

**Summary Of The Review:**

In brief, this paper presents an interesting few-shot AD method, which includes two steps: domain adaptation for pre-trained networks, and contrastive learning via building positive- and negative sample pairs. Evaluations are extensive and promising. However, the major concern is the novelty and its difference and comparison with the SOTA method such as CSI.

---

> ### Author Response · Authors · 2022-11-19
> **Response to Reviewer n46i**
>
> Dear reviewer, we highly appreciate the comments and suggestions on additional visualizations and comparison with state-of-the-art methods. In the response, we provide additional visualization of anomaly score distribution and comparison with existing methods. We hope the responses can ease the concerns from the reviewer and earn a full support to this work.
>
> ### It would be helpful to provide visualization after the outputs fit the multivariate Gaussian. This may further demonstrate if positive/negative pairs are grouped/separated as expected.
>
> We visualize the histogram of anomaly score for different methods in Appendix A.5 of the revised submission. Specifically, we compared the following methods, a) w/o contrastive finetuning; b) w/ contrastive finetuning; c) w/ PP loss; d) w/ NP loss. The figures are updated in the revised Appendix A.5. We make the following observations from the results. First, directly using the model pretrained on ImageNet, a) w/o contrastive finetuning, failed to differentiate normal and abnormal samples as the two distributions are almost the same. With contrastive finetuning on target data, b) w/ contrastive finetuning, we see a clear gap between normal and abnormal distributions, suggesting better anomaly detection performance. By further incorporating the positive pair loss, c) w/ PP loss, the gap between two distributions are more significant. Finally, incorporating negative pair loss, d) w/ NP loss, is most effective for differentiating normal from abnormal samples.
>
> ### Some experiment details are missing, e.g., how many normal images were used in Sec. 4.5.
>
> We used 10 normal images for training on CIFAR10-C and CIFAR100-C, i.e. 10-shot anomaly detection.
>
> ### Some competitive methods such as TDG (as mentioned in the paper) could be added to the experiments on cifar10-c and cifar100-c.
>
>
> We notice that TDG employed a GAN based approach thus the network architecture and anomaly inference paradigm is different from other methods which makes direct comparison unfair. Neverthelss, we still derive the numbers from the paper for comparison on few-shot anomaly detection on MVTec.
>
>
> ### The state-of-the-art CSI [1] can run under few-shot setting and could be evaluated here, too. Note most key steps in this paper are very similar to CSI, so it is strongly recommended to compare with CSI in few-shot tests.
>
> Thanks for referring to CSI. We notice that CSI adopted contrastive learning to learn representations that push augmented samples, referred to as "distribution shifted" samples, away from the original ones. This is similar to the negative pair loss introduced in our work. Consequently, CSI is theoretically less competitive for pretraining representation for distance based anomaly detection. We reproduced CSI for MVTec few-shot anomaly detection using the officially released code with results presented in the following table. Overall, the performance of CSI is much worse than our approach. This result is also updated in Table I of the revised manuscript.
>
> |             | bottle     | cable  | capsule | hazelnet | metal\_nut | pill  | screw | toothbrush |
> | ----------- | ---------- | ------ | ------- | -------- | ---------- | ----- | ----- | ---------- |
> | csi         | 80.55      | 60.48  | 62.07   | 74.40    | 59.83      | 69.29 | 32.71 | 79.44      |
> | ours w/o np | 94.58      | 70.67  | 74.06   | 78.29    | 79.66      | 81.56 | 62.70 | 87.18      |
> | ours w/ np  | 95.73      | 78.05  | 74.26   | 86.89    | 78.57      | 81.31 | 62.26 | 88.81      |
> |             | **transistor** | **zipper** | **carpet**  | **grid**     | **leather**    | **tile**  | **wood**  | **avg**        |
> | csi         | 55.71      | 63.71  | 56.17   | 39.53    | 51.56      | 55.38 | 75.96 | 61.12      |
> | ours w/o np | 78.08      | 75.03  | 84.67   | 62.31    | 74.90      | 78.34 | 89.66 | 78.11      |
> | ours w/ np  | 74.80      | 75.70  | 79.90   | 61.85    | 77.92      | 75.34 | 90.12 | 78.77      |
>
>
> ### It should be noted that in Fig. 1 after negative augmentation, the image however did not change to an anomaly, as expected.
>
> Thanks for the remind, we have revised the Fig.1 to reflect the true negative augmentation we carried out in the implementation.

---

### Official Review · Reviewer_gtEo · 2022-10-24

**Confidence:** 1
**Correctness:** 3
**Technical Novelty And Significance:** 1
**Empirical Novelty And Significance:** 2
**Recommendation:** 5

**Clarity, Quality, Novelty And Reproducibility:**

The paper is clear and well written overall. I don't see any huge problem with reproducibility

In terms of the problem, the datasets have a large number of defects in testing: usually in inference time anomaly detection tasks have much less anomalies than "normal" data. It would be interesting to see how the model behaves under a more "realistic", unbalanced, distribution.

All new ideas are somewhat already proposed before. Although the authors put together all those with success for the application at hand, that makes the paper less novel,

**Strength And Weaknesses:**

Strenghts:
- The approach shows good empirical results in 4 datasets
- The ideas for tweaking the learning using across distance positive pair and the negative pair loss may be useful in some scenarios beyond anomaly detection applied to industrial defect
- Many experiments are provided to investigate the behaviour of the method and the proposed ideas
- Using CutPaste as a way to synthesize negative instances is a good insight.

Weaknesses
- All new ideas are somewhat already proposed before. Although the authors put together all those with success for the application at hand, that makes the paper less novel
- The backbone may be deterrent to the final results. Maybe using a stronger backbone such as ViT-Dino could improve results of all baselines and diminish the amount of improvement claimed by the method.
- In terms of the application the paper does not provide a lot of insight. That is, why the proposed approaches benefit the application? Why does the defects are easier to spot after the representations are learnt with constrastive learning? Why this application is a good downstream task?
- The batchsize may play a significant role in the learning process, but this is not mentioned or studied in the paper.



**Summary Of The Paper:**

The paper proposes contrastive training and transfer learning applied to the problem of industrial defect identification via "anomaly detection". In particular, the negative pair loss is introduced as novelty in the paper.

**Summary Of The Review:**

The paper is well written and has some clever ideas, but there is no significant insight on the application, no reasons or insights are provided on why the methods work in the anomaly detection setting. Also, there is not much novelty on the main proposed methods.

---

> ### Author Response · Authors · 2022-11-19
> **Response to Reviewer gtEo part 2**
>
> ### The batchsize may play a significant role in the learning process, but this is not mentioned or studied in the paper.
>
> We use batchsize 64 throughout the experiments. Since we employ an exponential moving average model as the target model, there is no risk of trivial solution compared with SimCLR. So far we do not see much difference when tuning the batchsize.
>
> ### In terms of the problem, the datasets have a large number of defects in testing: usually in inference time anomaly detection tasks have much less anomalies than "normal" data. It would be interesting to see how the model behaves under a more "realistic", unbalanced, distribution.
>
> We notice that the original anomaly detection dataset is already highly imbalanced. For example, the AiTex dataset consists of 600 normal and 100 abnormal samples in the testing set. Since the ratio between normal and abnormal samples does not affect model training, we further implemented a controlled experiment where we manually decrease the number of anomalies in the SemiCon dataset. Specifically, we fix the normal samples to 500 and randomly subsample 5, 10, 20 and 50 abnormal samples for evaluation. The results in the follow table demonstrate the superiority of proposed method against existing competing methods regardless of the imbalance between normal and abnormal samples.  The above analysis has been updated in Appendix A.7.
>
> | #Anomalies | 5     | 10    | 20    | 50    |
> | ---------- | ----- | ----- | ----- | ----- |
> | deepsvdd   | 52.01 | 49.71 | 49.97 | 52.40  |
> | cutpaste   | 86.31 | 83.82 | 72.82 | 74.31 |
> | VAE        | 80.56 | 78.12 | 67.67 | 73.91 |
> | AE         | 62.12 | 65.52 | 76.83 | 74.39 |
> | ours       | 88.36 | 86.50  | 85.87 | 80.72 |

---

> > ### Comment · Reviewer_gtEo · 2022-12-07
> > **imbalancing**
> >
> > 6:1 is not a severe imbalance. Usually anomaly detection scenarios can reach up to a 100:1 imbalance. We do not expect more than 1% defects in production. Ideally this number should be even below 1%. We can see that the proposed approach, much more complex, improves only marginally cutpaste in the more severe imbalance scenario.

---

> ### Author Response · Authors · 2022-11-19
> **Response to Reviewer gtEo part 1**
>
> Dear reviewer, we highly appreciate the positive comments on extensive evaluations and introducing effective loss terms. We also appreciate the comments and suggestions on explaining the novelties, evaluating additional backbone networks and testing on imbalanced test data, which help us further improve the quality of this work. In the response, we provide experiments with improved backbone network and imbalanced test data. We hope the responses can ease the concerns from the reviewer and earn a full support to this work.
>
> ### All new ideas are somewhat already proposed before. Although the authors put together all those with success for the application at hand, that makes the paper less novel
>
> We believe identifying contrastive finetuning can help adapt pretrained model to target industrial image dataset and combining additional positive and negative pairs is not a trivial combination of existing techniques. Moreover, we identified that proper selections of data augmentation strategies for contrastive finetuning and negative pairs is essential to the success of representation learning for anomaly detection.
>
> ### The backbone may be deterrent to the final results. Maybe using a stronger backbone such as ViT-Dino could improve results of all baselines and diminish the amount of improvement claimed by the method.
>
> Thanks for suggesting to evaluate with stronger backbone network. In the revised manuscript, we further evaluated swapping ResNet18 with ResNet101 backbone which is compatible with our architecture. We believe contrastive training with transformer backbone should also be effective, but may require some hyperparameter tuning. In this experiment, we compared a contrastive baseline, contrastive with additional across instance positive pair and with additional negative pair. The results in the following table suggests better performance can be obtained with stronger backbone network and the proposed components are still effective with improved backbone network. The above results are updated in Appendix A. 6.
>
> | Contrast. Train | Positive Pair Loss | Avg. Acc. |
> | --------------- | ------------------ | --------- |
> | \-              | \-                 | 67.72     |
> | $\checkmark$     | \-                 | 84.22     |
> | $\checkmark$     | $\checkmark$        | 86.65     |
>
> ### In terms of the application the paper does not provide a lot of insight. That is, why the proposed approaches benefit the application? Why does the defects are easier to spot after the representations are learnt with constrastive learning? Why this application is a good downstream task?
>
> It has been adopted as a common practice for recent state-of-the-art anomaly detection approaches to initialize backbone network with ImageNet pretrained weights. We noticed that directly reusing backbone pretrained on natural semantic images is suboptimal for anomaly detection because there is a significant distribution shift between the source and target domain data. This motivates us to explore contrastive finetuning to adapt pretrained model to target domain data. This is further integrated with across instance positive pair and negative pair to improve the representation learning for downstream anomaly detection. We explain the success of contrastive finetuning for anomaly detection from the following two perspectives. First, we provide an analysis in the Appendix A.1. The backbone
> parameters $\Theta$ is optimized w.r.t. the source domain training data, e.g.
> the ImageNet dataset. Thus, $\Theta$ is suitable for differentiating semantic
> differences between images. For anomaly detection tasks, the image patterns
> feature a different distribution from semantic images. Contrastive training has
> been demonstrated to be adapt model parameters to target domain distribution
> [1] and we believe optimizing contrastive loss on target domain data, industrial
> images in this case, can help adapt model parameters to target domain. We further provide another perspective into the effectiveness of contrastive finetuning.
> When the augmentations are chosen to be mimic the commonly seen variations
> within normal samples, contrasting two augmented images forces the network
> to produce similar representations regardless of the augmentations. This means
> the representation learned from contrastive finetuning allows the network to learn
> features invariant to common variations in appearance and pose that one could
> encounter in industrial imaging environments. Such ability will help bring normal samples closer in the feature space, thus benefit downstream anomaly detection. Finally,  the specific task, i.e. few-shot anomaly detection on industrial images, motivates us to explore contrastive finetuning rather than the way around.

---

> > ### Comment · Reviewer_gtEo · 2022-12-07
> > **lack of novelty and contribution is shown for specific task/dataset**
> >
> > Thanks for the efforts in responding to the points. Although I believe the paper has merit, the techniques involved are all known (contrastive training, augmentation, positive/negative pairing) and were combined to the specific task and datasets explored in this paper. However the paper does not show sufficient evidence that this would generalize to a more broad range of applications, and in which conditions. Although the implementation is not trivial, overall I still believe the contributions are slightly below the acceptance threshold.

---

### Official Review · Reviewer_bC75 · 2022-10-26

**Confidence:** 4
**Correctness:** 3
**Technical Novelty And Significance:** 1
**Empirical Novelty And Significance:** 2
**Recommendation:** 3

**Clarity, Quality, Novelty And Reproducibility:**

Clarity:
As described in the previous section, the paper is not very clear in the presentation of the proposed method, especially some idea, notations, the network architecture.

Quality:
The paper still has some room for imprtovement interms of novelty, presentation and experimental justification.

Novelty:
Limited for the reasons given in the previous section.

Reproducibility:
Diifculty very reproduce since many details are missing in the paper.

**Strength And Weaknesses:**

Strengths:
1. The proposed idea of applying contrastive learning for adaptation seems interesting.
2. The experimental results on 4 different datasets are reported with promising results.

Weaknesses:
1. The paper is not well written. It is difficult to understand the main idea of the paper. How is the adaptation of a pre-trained model done with contrastive learning? Why doers the proposed training method provide good model adaptation? In addition, many important delaits for the implementation are missing. They didn't provide the network architecture of the proposed model.

2. The novelty proposed in this paper is limited. All the componenets used in this paper are not new. The idea of extending the contrastive training to model adaptation is not clear.

3. The notations used in describing the proposed method are not always defined or explained in the paper. For example, what are online encoder and target encoder in Figure 1 and eq. 2?

4. The experimental comparisons used in this paper should include comparison with more recent SOTA methods, such as
CFLOW-AD: Real-Time Unsupervised Anomaly Detection with Localization
via Conditional Normalizing Flows, WACV 2022
Registration based Few-Shot Anomaly Detection, ECCV 2022

5. The experiment protocols should follow the previous experiments published in the previous papers, such as those in Registration based Few-Shot Anomaly Detection, ECCV 2022, for a fair comparison.

6. For the denisity-based anomaly score for anomaly detection given in eq. (5), the covariance matrix could be signular if there is not enough samples in the convariance matrix estimation in eq. (4). There is no discussion on how to solve the problem.



**Summary Of The Paper:**

This paper is focused on the few-shot anomaly detection problem. It presents a method that combines contrastive learning for adapting a pre-training model to the target domain, a cross-instance positive pair loss, an option to incorporate negative pair loss, and density-based anomaly detection. They show some promising experimental results for few-shot anomaly detection on several public datasets.

**Summary Of The Review:**

The paper still has room for improvement in terms of paper presentation, technical novelty, and experimental justification. Please see the comments detailed in the previous sections.

---

> ### Author Response · Authors · 2022-11-19
> **Response to Reviewer bc75 part 2**
>
> ### 4. The experimental comparisons used in this paper should include comparison with more recent SOTA methods, such as CFLOW-AD: Real-Time Unsupervised Anomaly Detection with Localization via Conditional Normalizing Flows, WACV 2022
>
> We appreciate the suggestion to compare with some existing works. In the revised manuscript, we evaluated CFLOW-AD by pretraining on few-shot samples on MVTec dataset. The pretrained backbone is then used for downstream density based anomaly detection for fair comparison. We reproduced CFLOW-AD for few-shot anomaly detection on MVTec with the officially released code with results presented in the following table. Overall, CFLOW-AD is still behind our final approach. The results are updated in Tab. I of the revised manuscript.
>
> |             | bottle     | cable  | capsule | hazelnet | metal\_nut | pill  | screw | toothbrush |
> | ----------- | ---------- | ------ | ------- | -------- | ---------- | ----- | ----- | ---------- |
> | csi         | 80.55      | 60.48  | 62.07   | 74.40    | 59.83      | 69.29 | 32.71 | 79.44      |
> | cflow       | 98.17      | 81.65  | 73.12   | 88.25    | 74.88      | 68.22 | 45.09 | 82.50      |
> | ours w/o np | 94.58      | 70.67  | 74.06   | 78.29    | 79.66      | 81.56 | 62.70 | 87.18      |
> | ours w/ np  | 95.73      | 78.05  | 74.26   | 86.89    | 78.57      | 81.31 | 62.26 | 88.81      |
> |             | **transistor** | **zipper** | **carpet**  | **grid**     | **leather**    | **tile**  | **wood**  | **avg**        |
> | csi         | 55.71      | 63.71  | 56.17   | 39.53    | 51.56      | 55.38 | 75.96 | 61.12      |
> | cflow       | 84.79      | 83.53  | 73.48   | 50.96    | 87.84      | 91.59 | 92.37 | 78.43      |
> | ours w/o np | 78.08      | 75.03  | 84.67   | 62.31    | 74.90      | 78.34 | 89.66 | 78.11      |
> | ours w/ np  | 74.80      | 75.70  | 79.90   | 61.85    | 77.92      | 75.34 | 90.12 | 78.77      |
>
> ### 5. The experiment protocols should follow the previous experiments published in the previous papers, such as those in Registration based Few-Shot Anomaly Detection, ECCV 2022, for a fair comparison.
>
> Thanks for referring to this work. **The few-shot anomaly detection setting in RegAD is different from us and some recent few-shot works, e.g. TDG, which makes direct compairons with RegAD not applicable. The setting adopted by RegAD is also more restrictive.** As we discussed in the related work, RegAD (ECCV22) adopted a meta learning based few-shot approach. During training RegAD must iteratively update model on K-shot support set randomly selected from base classes. During testing, density model is fitted to the K-shot support set from novel class. In MVTec, the novel class refers to the class where only K-shot training samples are available while the base classes refer to all classes other than the novel class. In another words, RegAD rely on a large base class dataset, which is related to the novel class, for training. This is only applicable to certain scenarios where related industrial images are available, e.g. MVTec, and RegAD, and it can hardly generalize to more challenging scenarios where only a handful of images are available in the target domain, e.g. SemiCon, AiTex and Magnetic Tile datasets, evaluated in this paper.
>
>
> ### 6. For the denisity-based anomaly score for anomaly detection given in eq. (5), the covariance matrix could be signular if there is not enough samples in the convariance matrix estimation in eq. (4). There is no discussion on how to solve the problem.
>
> We adopt two approaches to tackle the lack of data issue. First, we implement data augmentation for K-shot training samples to expand the data samples $N_A=10$ times, as stated in Sect. 4.3 of the manuscript. The augmentation strategy is the same with the one adopted in contrastive training. In addition to data augmentation, as a common practice, we further add a small value on the diagonal of covariance matrix to reduce the condition number for improved numerical stability.

---

> ### Author Response · Authors · 2022-11-19
> **Response to Reviewer bc75 part 1**
>
> Dear reviewer, we highly appreciate the comments and suggestions on providing further insights into contrastive finetuning helps adaptation and comparison with state-of-the-art methods. In the response, we provide more detailed insights into the  mechanism of contrastive finetuning and additional comparison with state-of-the-art methods. We hope the responses can ease the concerns from the reviewer and earn a full support to this work.
>
>
> ### 1. The paper is not well written. It is difficult to understand the main idea of the paper. How is the adaptation of a pre-trained model done with contrastive learning? Why doers the proposed training method provide good model adaptation?
>
> To clarify, the pretrained model is finetuned using the contrastive loss on target domain data, which adapts the pretrained model's features to the target domain. This approach was shown to be effective for domain adaptation in [1].
>
> To elaborate, there are two reasons why our proposed method helps. First, we provide an analysis in the Appendix A.1. The backbone parameters $\Theta$ is optimized w.r.t. the source domain training data and task, (e.g. ImageNet dataset for classification). Thus, $\Theta$ is suitable for discriminating semantic differences between images. For anomaly detection tasks, the image patterns are from a different distribution from semantic images. Contrastive finetuning has been demonstrated to be adapt model parameters to the target domain distribution [1] and we believe by optimizing contrastive loss on target domain data (industrial images in this case) can help adapt model parameters to target domain. We further provide another perspective into the effectiveness of contrastive finetuning. When the augmentations are chosen to be mimic the commonly seen variations within normal samples, contrasting two augmented images forces the network to produce similar representations regardless of the augmentations. This means the representation learned from contrastive finetuning allows the network to learn features invariant to common variations in appearance and pose that one could encounter in industrial imaging environments. The learning procedure will help bring normal samples closer in the feature space, thus benefiting downstream anomaly detection. We have incorporated the above discussions in the Appendix A.1.
>
> [1] Liu Y, Kothari P, van Delft B, et al. TTT++: When does self-supervised test-time training fail or thrive?[J]. Advances in Neural Information Processing Systems, 2021, 34: 21808-21820.
>
> ### In addition, many important delaits for the implementation are missing. They didn't provide the network architecture of the proposed model.
>
> In principle, the proposed method is architecture independent. We additionally evaluated ResNet101 as the backbone network which demonstrates the effectiveness of the proposed approach. This can be found in Appendix A.6 in the revised submission. We shall release a demo for reproducing the results.
>
> ### 2. The novelty proposed in this paper is limited. All the componenets used in this paper are not new. The idea of extending the contrastive training to model adaptation is not clear.
>
> We would like to highlight the novel aspects of this work. First, we employed contrastive training on target domain data to allow better adaptation to the industrial image domain. We further incorporate across-instance-pair loss to further encourage representations that support density-based anomaly detection. Finally, we incorporate negative-pair loss when prior knowledge on potential anomalies is available.
>
> Our approach which combines contrastive training and learning metrics for is new for anomaly detection tasks. We also demonstrated that the choice of augmentation for different loss components play a huge role in the success of the method.
>
> ### 3. The notations used in describing the proposed method are not always defined or explained in the paper. For example, what are online encoder and target encoder in Figure 1 and eq. 2?
>
> Thanks for suggesting improving the description. We managed to improve the presentation of this paper. For online and target encoders, we add further clarifications in the revised manuscript. The online encoder refers to the network that is updated through back-propagation while the target encoder refers to the network that is updated through exponential moving averaging.

---

> ### Author Response · Authors · 2022-12-06
> **Willing to Address Further Concerns**
>
> Dear Reviewer bC75,
>
> We would like to thank for your valuable comments again. As per the responses, we have discussed the mechanism of contrastive training and provide insight into why contrastive training could help adapting pre-trained model to target domain distribution. We have included additional experiments with CFLOW-AD which again demonstrates the competitiveness of the proposed method. We also explained why a direct comparison with RegAD is not applicable as we adopted another realistic few-shot AD setting which has been used in earlier works [1].  Other comments and suggestions on additional evaluations, novelty and reproducibility are all discussed accordingly in the responses. We hope the response could ease your concerns and we are always ready to provide more explanations and/or evaluations.
>
> Best,
> Authors
>
> [1] Shelly Sheynin, Sagie Benaim, and Lior Wolf. A hierarchical transformation-discriminating generative model for few shot anomaly detection. In IEEE/CVF International Conference on Computer
> Vision, 2021.

---

### Official Review · Reviewer_LkYy · 2022-10-26

**Confidence:** 4
**Correctness:** 4
**Technical Novelty And Significance:** 2
**Empirical Novelty And Significance:** 3
**Recommendation:** 6

**Clarity, Quality, Novelty And Reproducibility:**

The paper is well readable. The theoretical novelty is limited, effectively showing the usefulness of the positive pair loss term. On the experimental side the use of datasets is a plus as well as the ablation study.

**Strength And Weaknesses:**

Strengths:
*Easy to read
*evaluation on industry datasets as opposed to simplistic standard image benchmarks
*Mostly an improvement over Cutpaste, showing as insight limited usefulness of negative examples as created by CutPaste for certain real datasets.
*Ablation study

weaknesses:
*straightforward extension with limited novelty
*limited usefulness of the second proposed loss term
*improvement over cutpaste on three datasets due to omission of the NP term, effectively demonstrating the gain due to the term encouraging clustering between positive samples.


**Summary Of The Paper:**

The paper considers outlier detection in a few-shot setting with a small samples size in the target domain.

They propose to combine self-supervised learning with two loss terms, one loss term to encourage larger similarities between in domain samples, and a second loss term to encourage smaller similarities between in domain samples and simulated outliers.

They use a mahalanobis distance using normalized features to score for outlierness, without using a mean term to subtract from the features.

They go ahead to demonstrate results on four industrial defect datasets and provide an ablation study on the terms used.


**Summary Of The Review:**

Effectively the paper shows the usefulness of self-supervised finetuning with an added positive pair loss.

---

> ### Author Response · Authors · 2022-11-19
> **Response to Reviewer LkYy**
>
> Dear reviewer, we highly appreciate the comments and suggestions on providing further insights and explanations on novelties. In the response, we provide more detailed insights into the usefulness of negative pair loss and novelties. We hope the responses can further reinforce your support for this work.
>
> ### Strengths: *Easy to read *evaluation on industry datasets as opposed to simplistic standard image benchmarks *Mostly an improvement over Cutpaste, showing as insight limited usefulness of negative examples as created by CutPaste for certain real datasets. *Ablation study
>
> We would like to thank the reviewer for appreciating the problem we addressed in this work, improvements over the state-of-the-art methods and presenting experiments on realistic anomaly detection tasks.
>
> ### *straightforward extension with limited novelty
> We wish to highlight that adapting contrastive learning for few-shot anomaly detection has not been done before, to the best of our knowledge. Our approach is motivated by the distribution shift between the source domain and target industrial images and we propose to introduce contrastive finetuning to reduce the impact of covariate shift. We further incorporated across-instance positive pairs to further encourage feature embeddings amenable to density-based anomaly detection.
>
> ### *limited usefulness of the second proposed loss term
>
> If the reviewer is referring to the negative-pair loss, we first highlight that using the negative-pair loss can result in good performance improvements. For instance, there is a 1.80% absolute improvement on AUROC in both 2-shot (68.06 $\rightarrow$ 69.86%) and 10-shot (80.92 $\rightarrow$ 82.71%) settings on the MVTec dataset. We also observe 2.30% absolute improvement on AUROC in the 10-shot setting on CIFAR10-C.
>
> We further characterized the conditions under which the negative-pair loss is helpful in Section 4.6.1 of the paper. Indeed, when prior knowledge is missing or synthesizing negative examples is not possible, the negative-pair loss is not helpful (e.g. on the SemiCon and AITEX datasets). However, when prior knowledge of anomalies is available and it is feasible to synthesize anomalies, we find that negative-pair loss can significantly improve performance as described above (e.g. on MVTec and Magnetic Tile datasets).
>
> ### *improvement over cutpaste on three datasets due to omission of the NP term, effectively demonstrating the gain due to the term encouraging clustering between positive samples.
>
> Indeed, our results without NP also validate the effectiveness of the across instance positive-pair loss, as also noted by the reviewer. Related to the previous point, we think that omitting NP improves performance because no appropriate prior knowledge to synthesize negative samples is available on these datasets. In fact, using CutPaste to synthesize negative samples is suboptimal for these datasets. We conclude that the negative pair loss should be used with caution whilst the limitation of used negative samples was not thoroughly discussed in the original CutPaste paper. That being said, when appropriate prior knowledge is available, the negative-pair loss can significantly improve performance as we demonstrated above.

---

> > ### Comment · Reviewer_LkYy · 2022-12-06
> > **reviewers opinion**
> >
> > The reviewer has read the comments, feels that the argument of being a straightforward extension still holds. It shows that the NP term in cutpaste is not so useful which has a certain value. The reviewer thinks that the original rating of the reviewer is justified.

---

### Author Response · Authors · 2022-11-19
**Response to All Reviewers**

We would like to highly appreciate all reviewers' efforts in providing valuable comments and constructive suggestions for improvement on our submission. First, we are very glad to see the positive comments by all reviewers. In particular, **introducing contrastive training for few-shot anomaly detection representation learning, addressing realistic anomaly detection tasks and competitive performance** are acknowledged by reviewers. In this response, **we shall address the questions over the novelty and insights of the proposed approach, comparisons with state-of-the-art methods, additional details and visualizations**. Specifically, we discussed why contrastive learning can help adapt models pretrained on ImageNet to target anomaly detection datasets from two perspectives. We also evaluated additional state-of-the-art anomaly detection methods under few-shot scenario. More details of the method and additional visualizations are presented for clarity and insights. Finally, we appreciate all reviewers again for considering this submission after revision for acceptance.

---

### Author Response · Authors · 2022-11-30
**Willing to Address Additional Comments**

Dear reviewers,

We would like to highly appreciate all reviewers' efforts and time again in providing valuable comments and constructive suggestions for improving our submission. We hope that the clarifications and additional evaluations provided in the responses have addressed all reviewers' questions and concerns.

We are always ready to provide additional clarifications should you have any questions and concerns during the discussion period.

Thank you very much!

Authors

---

### Decision · Program_Chairs · 2023-01-20

**Decision:**

Reject

**Justification For Why Not Higher Score:**

There is a concern remaining that the proposed approach is too straightforward of an application of existing ideas and too thin on technical novelty.

**Justification For Why Not Lower Score:**

N/A

**Metareview: Summary, Strengths And Weaknesses:**

The submission looks into the problem of few-shot anomaly detection. The approach it proposes first trains the model on a large source dataset and then uses a combination of contrastive learning (BYOL), cross-instance loss on pairs of positive examples, and a loss on pairs of positive and synthetic negative examples to adapt the model to the target domain. Anomaly detection is performed by fitting a multivariate Guassian distribution on the few positive examples, and the anomaly score is obtained through the Mahalanobis distance.

The proposed approach is evaluated on four industrial defect identification datasets (MVTec, AITEX, Magnetic Tile Defects, and SemiCon) and two datasets with synthetic common corruptions (CIFAR10-C, CIFAR100-C). It is compared against several baselines and is claimed to achieve state-of-the-art performance.

Reviewers generally feel positive about the submission's clarity; some reviewers raised questions on experimental details which were answered by the authors in the rebuttal. Most reviewers note the proposed approach's promising performance (bC75, gtEo, n46i). Some reviewers pointed out missing recent baselines (bC75 on CFLOW-AD, n46i on TDG and CSI), and the authors addressed this in the latest updated submission. Reviewers are split on acceptance: Reviewer LkYy is leaning towards acceptance, Reviewer n46i went from leaning towards rejection to leaning towards acceptance after the rebuttal, and Reviewer gtEo is leaning towards rejection.

The main outstanding concern shared by all reviewers is technical novelty. Reviewer LkYy points out that the proposed approach is a straightforward application of contrastive learning, but they also note that the observation on the uneven effectiveness of the negative pair term in CutPaste is of interest to the community. Reviewer gtEo is concerned that the submission is an application of existing ideas to a new problem. Reviewer n46i notes that the proposed approach shares design elements with CSI. In their response, the authors highlight the fact that contrastive learning has not been applied to few-shot anomaly detection before and clarify the similarities and differences of their proposed approach with CSI. Following the authors' response, Reviewers LkYy and gtEo remain concerned that the proposed approach is too straightforward of an application of existing ideas and too thin on technical novelty.

**Summary Of Ac-Reviewer Meeting:**

N/A